# Auto QSAR-based active learning docking for hit identification of potential inhibitors of *Plasmodium falciparum* Hsp90 as antimalarial agents

**Thato Matlhodi**[1], **Lisema Patrick Makatsela**[1], **Tendamudzimu Harmfree Dongola**[2], **Mthokozisi Blessing Cedric Simelane**[3], **Addmore Shonhai**[2], **Njabulo Joyfull Gumede**[4], **Fortunate Mokoena**[1] *

**1** Department of Biochemistry, Faculty of Natural and Agricultural Science, North-West University, Mmabatho, South Africa, **2** Department of Biochemistry & Microbiology, University of Venda, Thohoyandou, South Africa, **3** Department of Biochemistry, Faculty of Science, University of Johannesburg, Johannesburg, South Africa, **4** Department of Chemical and Physical Sciences, Faculty of Natural Sciences, Walter Sisulu University (WSU), Umthatha, Eastern Cape, South Africa

* Fortunate.Mokoena@nwu.ac.za

## Abstract

Malaria which is mainly caused by *Plasmodium falciparum* parasite remains a devastating public health concern, necessitating the need to develop new antimalarial agents. *P. falciparum* heat shock protein 90 (Hsp90), is indispensable for parasite survival and a promising drug target. Inhibitors targeting the ATP-binding pocket of the N-terminal domain have anti-*Plasmodium* effects. We proposed a *de novo* active learning (AL) driven method in tandem with docking to predict inhibitors with unique scaffolds and preferential selectivity towards PfHsp90. Reference compounds, predicted to bind PfHsp90 at the ATP-binding pocket and possessing anti-*Plasmodium* activities, were used to generate 10,000 unique derivatives and to build the Auto-quantitative structures activity relationships (QSAR) models. Glide docking was performed to predict the docking scores of the derivatives and > 15,000 compounds obtained from the ChEMBL database. Re-iterative training and testing of the models was performed until the optimum Kennel-based Partial Least Square (KPLS) regression model with a regression coefficient $R2 = 0.75$ for the training set and squared correlation prediction $Q2 = 0.62$ for the test set reached convergence. Rescoring using induced fit docking and molecular dynamics simulations enabled us to prioritize 15 ATP/ADP-like design ideas for purchase. The compounds exerted moderate activity towards *P. falciparum* NF54 strain with $IC_{50}$ values of $\leq 6\mu M$ and displayed moderate to weak affinity towards PfHsp90 ($K_D$ range: 13.5–19.9μM) comparable to the reported affinity of ADP. The most potent compound was FTN-T5 (PfN54 $IC_{50}$:1.44μM; HepG2/CHO cells SI$\geq$ 29) which bound to PfHsp90 with moderate affinity ($K_D$:7.7μM), providing a starting point for optimization efforts. Our work demonstrates the great utility of AL for the rapid identification of novel molecules for drug discovery (i.e., hit identification). The potency of FTN-T5 will be critical for designing species-selective inhibitors towards developing more efficient agents against malaria.

**Data Availability Statement:** All relevant data are within the manuscript and its Supporting information files.

**Funding:** FM was awarded the Grand challenges Africa drug discovery seed grant (GCA/Round10/DD-065), funded by the Bill and Melinda Gates foundation is hereby acknowledged. "The funders had no role in study design, data collection and analysis, decision to publish, or preparation of the manuscript"".

**Competing interests:** The authors have declared that no competing interests exist.

# Introduction

Sub-Saharan Africa, especially marginalized populations, has recorded over 90% of the 608,000 deaths caused by malaria in 2022 [1]. Despite the presence of a malaria vaccine, the emergence of resistant strains indicates a threat to the gains made from decades of implementing malaria control strategies [2, 3]. It has also been suggested that changes in climate conditions such as increased temperatures and heavy rainfall may result in an increased mosquito population, putting more people at risk of contracting malaria [1]. Countries such as Rwanda [4] and East Asia [5] have begun to report the spread and dissemination of first-line treatment options artemisinin-tolerant *P. falciparum* strains emphasizing the urgent need to develop potent and reliable anti-parasitic drugs. Future antimalarials should inhibit *Plasmodium* infection and growth, potentially counteracting the likelihood of rapid development of drug resistance. Innovative approaches could explore validated drug target proteins implicated in drug resistance, such as PfHsp90 [6].

In *P. falciparum*, Hsp90 plays a crucial role during parasite adaptation and development, from the vector and host environment, which are often accompanied by abrupt increases in temperature amongst other stresses [7, 8]. PfHsp90 is expressed and essential for the parasite's survival at all erythrocytic [9–12] and hepatic stages of development [13]. Distinct expression profiles of PfHsp90 have been correlated to poor disease prognosis in *P. falciparum*-infected individuals [14], making it a prime drug target. PfHsp90 is a dimeric protein composing of the N-terminal domain (NTD), middle domain and c-terminal domains respectively serving as binding sites of ATP, client proteins and co-chaperones [15, 16]. Most Hsp90 inhibitors are small molecules, which compete with ATP for binding the NTD. In the literature these small molecules, including geldanamycin (GDA), 17-AAG, 17-DMAG and PUH-71, were shown to be effective antimalarial agents by inhibiting the activities of PfHsp90 [17–19]. Treatment of parasite cultures with GDA prevented their growth from the late ring to trophozoite stages of development [17]. *In vivo* studies demonstrated that in *P. berghei*-infected mice models, parasite load was reduced by treatment with GDA [17] and harmine [20]. PfHsp90 was identified as one of the artemisinin-based combination therapies' (ACT's) resistance-conferring alleles [6] and has been suggested to interact directly with chloroquine resistance transporter (CRT) protein [18]. Thus, targeting PfHsp90 for malaria treatment could be highly profitable as conserved proteins are less prone to variation under selection pressure, possibly overcoming the hurdle of drug resistance [21].

The high sequence conservation between the druggable ATP binding pocket of PfHsp90 versus human Hsp90 poses a risk with regards to human Hsp90 off-target activity. However, harmine was demonstrated to interact selectively with PfHsp90 using Arg98, which is substituted with Lys112 in human Hsp90 [20]. Following this discovery, a combination of rational design and microwave-assisted synthesis were used to derive several analogues of harmine [22, 23], leading the observation thattetrahydro-β-carboline possess moderate activity against *P. falciparum* [22, 24]. Comparative structural analyses conducted by Wang and colleagues (2014) revealed that a glycine hinge loop lining, found in the NTD of both chaperones, adopt different conformations. In PfHsp90, these residues include Gly118, Gly121, and Gly123 [25] adopting a straight conformation which enables better accommodation of some compounds. Therefore, selectivity towards PfHsp90 can be obtained by targeting the Arg98 interaction and cushioning especially hydrophobic segments of compounds into the glycine rich region [25]. These observations led Wang and colleagues (2016) to use structure guided design to identify amino alcohol-carbazoles as selective inhibitors of PfHsp90 [26]. Since then, other studies have also suggested new compound scaffolds displaying anti-*Plasmodium* activity through targeting PfHsp90 [27, 28]. We have recently used pharmacophore models to suggest four

chemically diverse inhibitors targeting PfHsp90 [29], suggesting the benefit of rational design in drug discovery efforts aimed at selective inhibition of the molecular chaperone.

Artificial intelligence and machine learning/active learning (ML/AL)-based virtual screening methods have proven effective in designing candidate compounds that have advanced to clinical trials.ls [30]. A selective A2A receptor antagonist, for instance, is currently undergoing phase 1b/2 studies to be used in patients with solid tumors with elevated adenosine levels [29]. To identify hits and lead compounds, quantitative structure-activity relationship models, or Auto-QSAR, have been employed [30]. Auto-QSAR is essentially an application of artificial intelligence and machine learning. Previous studies have implemented auto-QSAR to identify candidate hits for the etiology of Alzheimer's disease (AD) [33] and and Chagas disease [31].

Furthermore, the utility of AL models that incorporate docking scores in streamlining the drug discovery process have been demonstrated. In a recent study, regression-based AL models were used to rapidly prioritize compounds in large-scale docking, enhancing efficiency and reducing costs [31]. AL models can effectively select promising candidates for experimental validation and contribute to accelerating the development of therapeutic agents. These instances underline the potential of docking-based AL models in drug discovery. By leveraging the predictive capabilities of docking scores within AL frameworks, researchers can enhance the drug discovery pipeline's speed and efficacy.

With this in mind, we aimed to use AL models to generate new chemical entities targeting PfHsp90. As such, compound 10 (S1B Fig in S1 File) from the literature [28] was used as a reference compound; its choice was motivated by its demonstrated potency towards *P. falciparum* and low cytotoxicity towards human cells. Optimization of compound 10 was undertaken by reaction-based enumeration to generate AL models enriched by molecular docking. The models were trained using analogues of compound 10 and a subset of *de novo* compounds were docked against PfHsp90 and re-iteratively tested for activity. Candidate hit compounds generated were evaluated for whole cell potency towards *P. falciparum* NF54 drug-sensitive strain, and the safety profiles of the compounds were established by cytotoxicity assays using mammalian Chinese hamster cells (CHO) and human hepatocellular carcinoma (HepG2). Select compounds which exhibited anti-plasmodial activity at the asexual blood stages were then evaluated for their binding affinity towards PfHsp90.

## Methods

### *In silico* analyses

Schrödinger Release 2022–1 was used for all molecular modelling calculations on Maestro (v12.9) [32], as a graphical user interface (GUI). Several modules in Maestro for ligand preparation, protein preparation, docking, QSAR modelling and docking post-processing were used. The Fig 1 shows the *in silico* process flow-chart of the steps undertaken to design and generate potential novel inhibitors of PfHsp90.

### Protein preparation and receptor grid generation

The three-dimensional (3D) structures of the NTDs of PfHsp90 bound to ADP (PDB code: 3K60-2.3 Å; Chain A; [33], human Hsp90 in complex with geldanamycin (GDA) (PDB code: 1YET-1.9 Å; [34]) and human Hsp90 bound to ADP (PDB code: 1YBQ-1.5 Å; [35]) were obtained from the protein data bank (http://www.rcsb.org/; [36]). The protein preparation wizard module of Schrödinger Maestro [37] was used to add hydrogen atoms, correct bond orders, disulphide bonds, filling in of missing/incorrect side chains and loops of each protein. The system was minimized with Root-Mean-Square Deviation (RMSD) convergence of 0.30Å using an optimized potential for liquid simulations 4 (OPLS4) force field [32]. Grid files at the

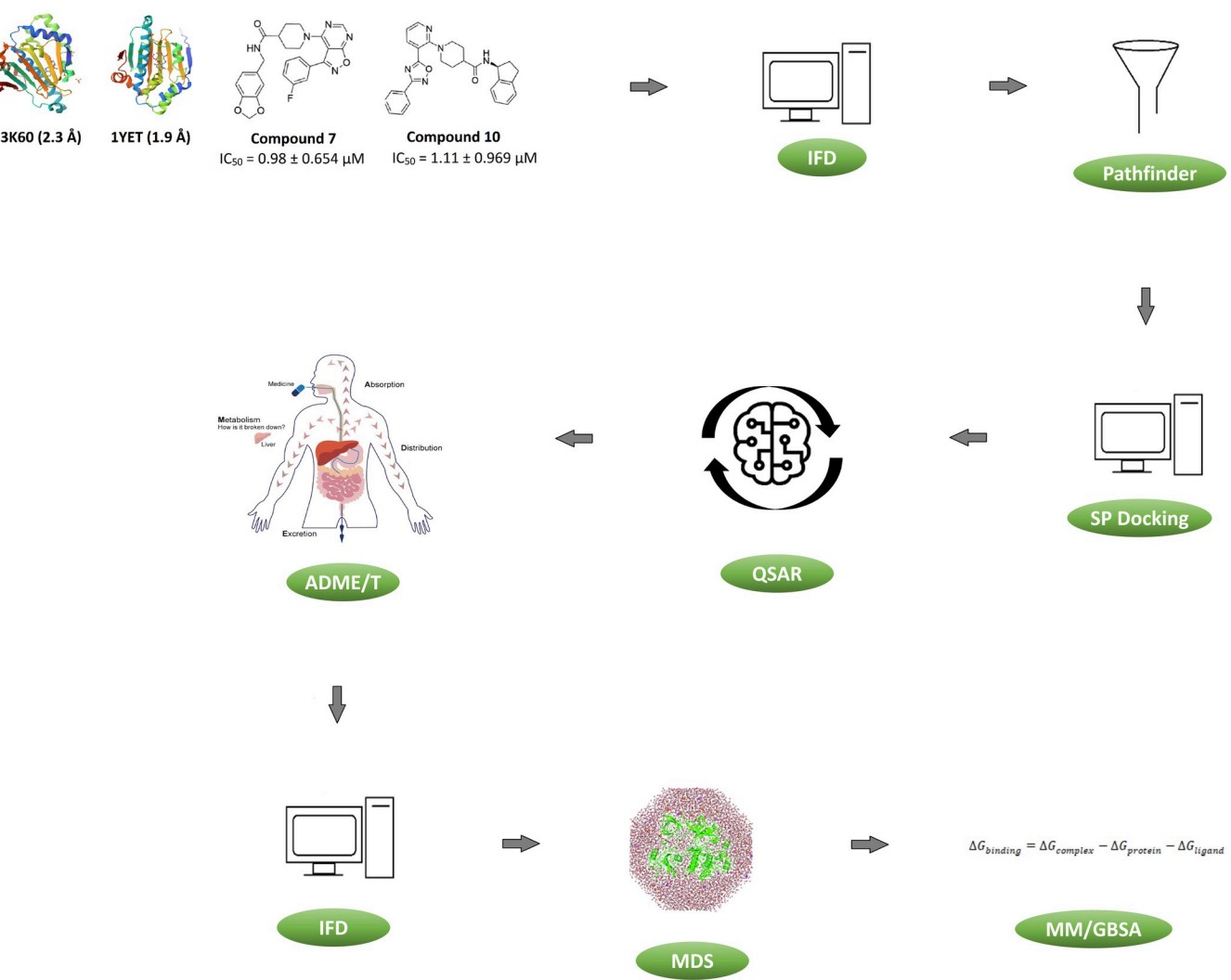

**Fig 1. *In silico* flow diagram showing methodology used on how the reference compounds 10 and 7 were subjected to induced fit docking (IFD) using human Hsp90 (PDB code: 1YET) and PfHsp90 (PDB code: 3K60).** Followed by the selection of compound **10** which was subsequently subjected to pathfinder reaction-based enumeration to yield 10, 000 unique design ideas. These design ideas and compounds from the ChEMBL database were further used to train 13 Auto-QSAR models through reiterative training and testing and scoring of the compounds using standard precision (SP) Glide docking. The adsorption, distribution, metabolism, excretion, and toxicity (ADME/T) were used to filter the compounds, followed by rescoring by IFD and molecular dynamics simulations (MDS) to understand the stability of the protein-ligand complex and conformational changes induced upon ligand binding. The relative binding free energies were estimated by molecular mechanics with generalized born surface area (MM-GB/SA).

centroid of ADP/GDA were generated using Receptor Grid Generation module for subsequent Glide docking [38].

### Ligand preparation

In a previous study conducted by Everson and colleagues (2021), compounds 7 (*P. falciparum* 3D7a $IC_{50}$ = 0.98 ± 0.654 μM) and compound 10 (*P. falciparum* D7a I $C_{50}$ = 1.11 ± 0.969 μM) were shown to be near sub-molar potency towards chloroquine-sensitive *P. falciparum* strain. Due to their proven potency, these compounds were then used as reference compounds for this study. Therefore, the 2D structure of each compound was drawn using 2D sketcher on

Maestro and converted to low-energy 3D structures with tautomeric states using the Ligprep module [39]. The input structures were optimized using OPLS4 force field, generating possible ionization states at a target pH of 7.4 +/- 2.0 using Epik. Stereoisomers were computed to retain specified chiralities at most, generating 32 per ligand [39]. This procedure was repeated for the >2 million compounds from ChEMBL database (https://www.ebi.ac.uk/chembl/g/ #browse/compounds) and 10 000 enumerated products.

## Induced fit docking (IFD)

To predict the binding modes and nature of interaction in PfHsp90 upon binding different ligand, Induced Fit Docking (IFD) was implemented as described by [29]. This study implemented IFD for two main purposes: the first was to dock compound 7 and 10 to the ATP binding pocket of PfHsp90, to understand their affinity and mode of binding; and secondly, to rescore newly generated ligands following Auto QSAR prediction. Therefore, a grid box of the binding site for the prepared structure of PfHsp90-NTD was generated considering the co-crystallized ligand ADP as a centroid. This was followed by the removal of ADP from ATP binding region to provide more room for ligand docking and specifying Asn37, Arg98 and Phe124 as binding residues [33]. To refine the side chains of residues located within a 5Å distance from the ligand, the Prime refinement step was employed. After the initial docking of each ligand, up to 20 poses were selected for further refinement using the XP mode. The best pose of each complex was selected based on docking scores, and a visual inspection of the binding orientation. The visual inspection involved assessing the residues of the protein involved in binding and orientation of the ligand. Most importantly, assessing the presence of a hydrogen bond between PfHsp90 and the residue, Arg98, which confers selectivity.

## Pathfinder reaction-based enumeration

Compound 10 exhibited a better fit into the ATP bin ding pocket of PfHsp90 following induced fit docking analysis. The possible routes to synthesize compound 10 were predicted by retrosynthesis analysis using Pathfinder [40]. Briefly, the 2D structure of compound 10 was minimized and the lowest energy conformer was determined using the macro model module [40]. The regiochemistry of bonds that can be disconnected were displayed with a maximum depth set to 1. Possible retrosynthesis pathways were estimated by employing Pathfinder, revealing nine pathways for the coupling reactions that are possible for the synthesis of compound 10. Pathways 1 and 2 were Amide_coupling-1 & 2, pathway 3 was amination-1, pathway 4- was Hayima-1, pathway 5 was Negishi, pathway 6 was oxadiazole-1, pathway 7- was Stille, and finally, pathways 8 and 9 were Suzuki-1 and-2 cross-coupling reactions. The enumeration reaction was employed by using pathway 6. The oxadiazole-1 pathway was chosen due to its synthetic accessibility and favourable reaction conditions, which could result in a higher success rate in generating diverse compound structures [41]. We surmised thatdesign idea compounds from pathway 6 woulddemonstrate chemical features that would have more affinity and more conducive to interaction with the PfHsp90 receptor. To do this, the reactants were defined, with reactant 1 containing the benzonitrile moiety was varied by enumerating nitrile fragments from the e-molecules database. The core/original reactant for reactant 2 was retained (Fig 2). Since, this core contains important functional groups for recognition of PfHsp90. Default physiochemical parameters such as a molecular weight (MW) between 150 and 575g/mol, LogP between −1.50 and 5.0, a topological polar surface area (TPSA) between 30 and 150, HBA between 0 and 12, and HBD between 0 and 5, and a maximum number of rotatable bonds less than 10 were retained for the design ideas [42]. Reactive functional groups using smiles arbitrary target specification (SMARTS) and Pan Assay Interfering Structures

## Oxadiazole-1 Pathway

**Fig 2. Pathway 6 which is oxadiazole-1 coupling was chosen as our pathway of interest.** Reactant 1 showing the benzonitrile moiety, whilst reactant 2 depicting the core structure containing the pyridine-3-carboxylic acid moiety.

(PAINS) offenders were removed [43]. This enumeration round resulted in 10 000 design ideas that needed to be further tested for their binding affinities to PfHsp90.

### ChEMBL database and enumerated compound preparation

The chemical space was enriched by adding approximately 15 000 randomly selected compounds from the >2 million compounds from ChEMBL database (https://www.ebi.ac.uk/chembl/g/#browse/compounds). A total of 10 cycles of GlideSP docking was conducted in each round compounds from ChEMBL and the enumerated ideas were randomly selected and prepared for docking.

### Classical glide SP ligand docking

Glide-based workflow have previously been used to screen many compounds against a target receptor quickly and accurately [38]. Glide energy terms offer SP (standard precision) for reliably docking ligands with high accuracy, or XP (extra precision) mode, which further eliminates false positives by extensive sampling and advanced scoring, resulting in high enrichment [38]. In this study, the two sets of ligands (1000 enumerated design ideas and 1000 ChEMBL database ligands for the initial GlideSP docking) were subjected to ligand docking studies to select ligands exhibiting favourable binding affinity ($\geq$ -5 kcal/mol) towards PfHsp90. The ligands exhibiting favourable binding energies were then used to build the first active learning (AL) model.

### Receptor grid generation

A receptor grid was generated based on ligand binding residues to specify the position and size of the receptors active site for glide SP ligand docking, utilizing the receptor grid generation

tool in Maestro v12.9. ADP was decoupled from the receptor and positional constraints were defined as Asn37, Arg98 and Ile173 in PfHsp90 [25, 33].

## Auto QSAR-AL models enriched by GlideSP docking

Auto quantitative structure activity relationship (QSAR) is a best practice protocol for generating models with limited user input and understanding. It also builds categorical or numerical models based on physicochemical, topological descriptors and binary fingerprints (i.e., radial, linear, dendritic, and 2-D molecular prints) [42], where a given model is trained against a particular random subset of input structures [44]. This study used the Auto QSAR to construct the AL models. The initial and second models were built by utilizing ligands exhibiting docking scores $\geq$ -5 kcal/mol. Subsequently, models 3 to 9 were constructed employing ligands characterized by docking scores of $\geq$-6.0 kcal/mol. Models 10 and 11 were developed specifically with ligands demonstrating docking scores $\geq$-6.5 kcal/mol. Lastly, models 12 and 13 were built using ligands possessing docking scores of $\geq$-7 kcal/mol. In each iterative cycle, the ligands that were chosen in prior rounds were added with newly selected ligands. These combined ligand sets were then partitioned into a training subset encompassing 75% of the data and a test subset comprising the remaining 25%. Internal validation of the model was assessed using prediction parameters such as predictive precision of the root mean square error (RMSE), standard deviation (SD), the accuracy of the training set (R2) and lastly the accuracy of the test set (Q2) in order to rank of all models [44]. The models were trained, tested, and retrained using ligands from enumerated designs from pathway 6 and the ChEMBL dataset until the model reached convergence. To assess the performance of model 13 Mean Absolute Error (MAE) analysis was performed by measuring errors between paired observations (predicted activity and observed activity).

## Molecular dynamics simulations

Select docking poses of PfHsp90-inhibitor complexes were subjected to Molecular Dynamics Simulations using the Desmond package on Maestro [45]. The complexes were solvated in transferable intermolecular potential with 3 points (TIP3P) water model and enclosed into orthorhombic boxes with minimized volumes. Ions were added to neutralize the charges using force field OPLS4. Then, the simulation was conducted as described by [46] with minor adjustments. In general, the simulation was allowed to proceed for 50–150 ns, 100 ps trajectories and 1000 frames. The NPT ensemble class was selected at a temperature of 300 K and a pressure of 1.01 bar.

## Free binding energy calculations

The last trajectory from molecular dynamics simulation was subjected to molecular mechanics with generalized born surface area (MM-GBSA) to establish the free binding energies contributing to the protein-ligand interactions. In this study, employing the prime module in Maestro (v13.2) from the Schrödinger Suite 2022–1, the molecular mechanics with generalized born surface area (MM-GBSA) was performed. This was conducted to calculate free binding energies to determine the stability of the protein-ligand complexes from the docking conformations. MM-GBSA estimates the binding free energy of a ligand-receptor complex by combining molecular mechanics force fields, which describe the intermolecular interactions, with a solvation model based on the Generalized Born (GB) theory [47].

The calculations for the docked complexes were subjected to an OPLS4 force field, a VSGB solvation model and the sampling was minimized. The free binding energies were therefore

calculated using the following equations:

$$\Delta G_{bind} = \Delta E + \Delta G_{solv} + \Delta GSA \tag{1}$$

$\Delta E$ denotes the difference in minimized complex energy and $\Sigma$ energies of unbound receptor and ligand. $\Delta G_{solv}$ is the difference in GBSA solvation energy of receptor-ligand complex and $\Sigma$ solvation energies of unbound receptor and ligand. $\Delta GSA$ represents the difference in surface area energy of IFD complex and $\Sigma$ surface area energies of unbound receptor and ligand.

$$\Delta E = E_{complex} - E_{protein} - E_{ligand} \tag{2}$$

$E_{complex}$, $E_{protein}$, and $E_{ligand}$ represent the minimized energies for the protein-inhibitor complex, the protein, and inhibitor, respectively.

$$\Delta G_{solv} = \Delta G_{solv\ (complex)} - \Delta G_{solv\ (protein)} - \Delta G_{solv\ (ligand)} \tag{3}$$

$$\Delta GSA = \Delta GSA_{(complex)} - \Delta GSA_{(protein)} - \Delta GSA_{(ligand)} \tag{4}$$

$\Delta GSA$ is the nonpolar contribution to the solvent energy of the surface zone. $GSA_{(complex)}$, $GSA_{(protein)}$, and $GSA_{(ligand)}$ denote the surface energies of the protein-inhibitor complex, the protein, and the ligand, respectively.

### *In vitro* methods

**Reagents.** Unless otherwise stated, all reagents were purchased from Thermo Fischer Scientific (USA), Sigma-Aldrich (USA) and Promega (Madison, USA).

### *In vitro* anti-*Plasmodium* assay

*In vitro* anti-*Plasmodial* assays were conducted at the H3D testing centre at the University of Cape Town. All 15 commercially available compounds were evaluated for anti-*Plasmodium* activity using a parasite lactate dehydrogenase assay as a marker for parasite survival, as previously described by [48]. Briefly, the parasites were synchronized at the ring stage using d-sorbitol in water. Approximately 2 mg/mL stock solutions of reference drugs, chloroquine and artesunate, were prepared in water and DMSO respectively then stored in -20˚C. Test compounds and reference drugs were serially diluted to give 10 concentrations with a final volume of 100 µL in each well. Parasites were incubated in the presence of the compounds at 37˚C under hypoxic conditions (4% CO2 and 3% O2 in N2) for 72 h. The absorbance was measured at 620 nm on a microplate reader. Survival was plotted against concentration and the $IC_{50}$-values were obtained using a non-linear dose-response curve fitting analysis via the Dotmatics software platform.

### *In vitro* cytotoxicity

Selectivity of the compounds for the parasites was determined using two cytotoxicity assays both of which were conducted at the H3D Centre. Compounds were screened against the mammalian cells, Chinese Hamster Ovarian (CHO) and hepatocellular carcinoma (HepG2), using the 3-(4,5-dimethylthiazol-2-yl)-2,5-diphenyltetrazoliumbromide (MTT) assay [49]. Cells were plated to a density of 105 cells/well in 96-well plates and allowed to attach for 24h. After that compounds were added at various concentrations from 50µM to 16nM and the cells incubated for a further 48h. Emetine was used as the control. After that MTT was added and plates were read 4h later. "Survival" was plotted against concentration and the $IC_{50}$-values

were obtained using a non-linear dose-response curve fitting analysis via the Dotmatics software platform.

### Expression and purification of PfHsp90-NTD and surface plasmon resonance

The NTD of PfHsp90 was expressed and purified as described in [29] and 20μg/ml of the protein was used to study the binding affinities of selected ligands using BioNavis Navi 420A ILVES multi-parametric surface plasmon resonance (MP-SPR) system (BioNavis, Finland) as previously described [50]. Briefly, PfHsp90-NTD was immobilized to a Carboxymethyl dextran coated sensor slide (BioNavis SPR102- CMD-3D). The sensor slide was activated with 0.1 M EDC/0.05 M NHS. At pH 5, PfHsp90-NTD suspended in 5 mM of sodium acetate was immobilized on one channel of the 3 CMD chip. 1 M ethanolamine HCl was used to deactivate the chip by removing excess NHS = EDC, and the sensor cleaned with NaCl/NaOH unspecific binding molecules. Varying concentrations (0-2000/5000 nM) of compounds FTN-T2 and FTN-T5 were injected and introduced to the flow cell, allowing binding to the surface. Data Viewer (BioNavis, Finland) and Trace Drawer software version 1.8 (Ridgeview instruments, Uppsala, Sweden) were used to process and analyze the steady-state equilibrium constant data and estimate the binding affinities.

## Results

### Induced fit docking (IFD) of reference compounds

Previous studies have demonstrated the benefit of rational strategies in designing selective inhibitors of PfHsp90 [26, 28, 29]. We used compounds 7 and 10 from Everson et al., 2021 which were shown to have potency towards *P. falciparum* drug-sensitive strain while exhibiting a safety profile towards mammalian cells. This study sought to generate novel inhibitors with unique scaffolds and preferential selectivity towards binding PfHsp90, a validated malaria drug target. The main challenge with selective targeting of PfHsp90 is the correspondingly high sequence and structural similarity of the ATP/druggable pocket of the chaperone versus that of human Hsp90. Compounds concomitantly inhibiting PfHsp90 and human Hsp90 would likely result in unintended toxicity.

The IFD results showed compound 7 had a slightly lower docking score of -8.031 kcal/mol (data not shown), making fewer contacts with the PfHsp90 binding pocket. Only two interactions were observed, a hydrogen bond with Phe124 and a water-mediated hydrogen bond with Lys44 (Fig 3). Compound 10, conversely, had a better fitting into the ATP binding pocket of PfHsp90, evidenced by a docking score of -10.485 kcal/mol (data not shown). Compound 10 made several hydrogen water-mediated bond interactions with amino acid residues such as Lys44 and Arg98, respectively and pi-pi interactions with Phe124 and Trp148 (Fig 3). It was surmised that the interaction with Arg98 could be the basis for the selectivity of the compound as has been previously described for other selective inhibitors of PfHsp90 [18, 20].

The PfHsp90-compound **10** complex was examined for functional groups forming important interactions. It was seen that the phenyl ring of the 2-phenyl-1,3,4-oxadiazole moiety did not participate in hydrogen bonding or π-πinteraction. Therefore, the phenyl ring was disconnected from the 2-phenyl-1,3,4-oxadiazole moiety (Fig 2), giving the synthetic precursors such as benzonitrile (reactant 1) and the core-containing pyridine-3-carboxylic acid (reactant 2). Here, the intention was to generate compounds with better affinity for PfHsp90 than compound **10** without altering the core structure.

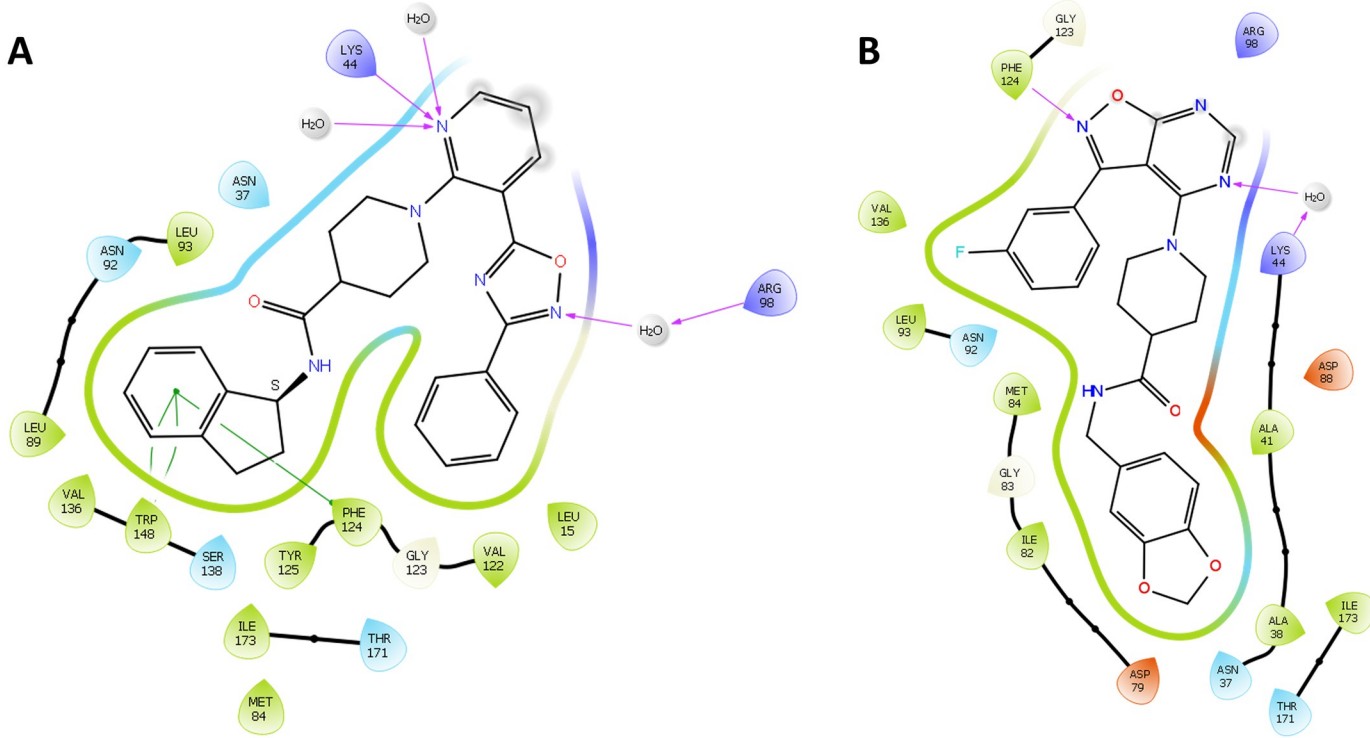

**Fig 3. Details the ligand interaction diagrams for compound 10 and 7 in the PfHsp90 N-terminal domain ATP binding site. A**. Compound **10** making pi stacking (shown by green lines) interactions with Phe124 and Trp148, a water mediated hydrogen bond with Arg98 and lastly a direct hydrogen bond with Lys44. **B.** Compound **7** making a water mediated hydrogen bond with Lys44 and a direct hydrogen bond with Phe124. Positively charged amino acids indicated in blue, negatively charged amino acids in orange, polar in light blue, non-polar in green, and hydrogen bond interactions in purple.

### Pathfinder analysis and reaction-based enumeration of compound 10

To generate analogues of compound 10, incorporating unique fragment and with probably more affinity for PfHsp90, we used Pathfinder reaction-based enumeration to create an extensive library of synthetically tractable compounds *in silico* as was conducted by Konze and colleagues (2019). Approximately 10, 000 design ideas of enumerated products incorporating unique drug-like fragments from commercial databases in place of reactant 2 of compound 10 were generated.

### AutoQSAR models

Table 1 details the 13 models built using the AL approach including their statistical parameters. In each round of model building, parameters such as the values of the ranking score of the model (score) being the standard deviation of the model (SD), the training set accuracy ($R2$), root-mean square error of the test set predictions (RMSE), and test set accuracy ($Q2$) were used to select the best performing model. A value of 1 in $R2$ and $Q2$ indicate a perfect prediction and a value of 0 for SD and RMSE indicate accuracy. Table 1 indicates the parameters for evaluating each model's performance (overall score, $R2$, $Q2$, RMSE and SD) and the total number of compounds in the training/test used to build the model. For example, the first model was built by randomly selecting 2000 compounds (1000 enumerated design ideas and 1000 from ChEMBL) aswell as compound 7 and 10. Ligands with docking scores ranging from -5.0 kcal/mol to -7.6 kcal/mol were retrieved yielding 123 ligands that satisfy the criteria

**Table 1. Statistical data for all 13 AutoQSAR prediction models.**

| Model | Model code | Score | S.D. | $R^2$ | RMSE | $Q^2$ | Training set | Test set |
|---|---|---|---|---|---|---|---|---|
| 1 | Kpls_linear_36 | 0.75 | 0.37 | 0.77 | 0.36 | 0.67 | 92 | 31 |
| 2 | Kpls_molprint2D_31 | 0.64 | 0.45 | 0.63 | 0.42 | 0.54 | 141 | 47 |
| 3 | Kpls_radial_34 | 0.65 | 0.46 | 0.68 | 0.43 | 0.59 | 161 | 54 |
| 4 | Kpls_molprint2D_42 | 0.58 | 0.44 | 0.68 | 0.47 | 0.53 | 170 | 57 |
| 5 | Kpls_radial_37 | 0.67 | 0.46 | 0.67 | 0.44 | 0.60 | 190 | 64 |
| 6 | Kpls_desc_37 | 0.60 | 0.48 | 0.59 | 0.45 | 0.59 | 210 | 71 |
| 7 | Kpls_desc_44 | 0.54 | 0.47 | 0.59 | 0.48 | 0.53 | 231 | 78 |
| 8 | Kpls_molprint2D_27 | 0.59 | 0.52 | 0.58 | 0.49 | 0.60 | 262 | 88 |
| 9 | Kpls_desc_33 | 0.59 | 0.45 | 0.71 | 0.50 | 0.60 | 290 | 97 |
| 10 | Kpls_radial_33 | 0.63 | 0.46 | 0.71 | 0.49 | 0.62 | 303 | 101 |
| 11 | Kpls_radial_36 | 0.58 | 0.52 | 0.63 | 0.53 | 0.57 | 317 | 106 |
| 12 | Kpls_radial_42 | 0.60 | 0.55 | 0.64 | 0.56 | 0.58 | 323 | 108 |
| 13 | Kpls_desc_23 | 0.56 | 0.48 | 0.75 | 0.56 | 0.62 | 342 | 114 |

($\geq$ -5.0 kcal/mol), including the two reference compounds. These high-affinity compounds were then employed to train Auto QSAR models, utilizing an AL approach. AutoQSAR divided the selected ligands into the training and test sets, with a random training set at 75% (92 compounds) and test set at 25% (31 compounds), the protocol was repeated for 13 models. MAE results of our model ranges from 1 to -1, giving an indication that the predictions are quite accurate as the closer MAE is to 0, the more accurate the model is.

It can be noted on Table 1 that all the models were generated using kernel-based partial least square regression (KPLS) differing in terms of binary fingerprints. Model 1 used linear, models 2,4 and 8 used molprint2D, models 3, 5, 10–12 used radial and the remaining models used KPLS descriptions binary fingerprints (Table 1). It is possible that the difference in the binary fingerprints of the top scoring models was caused by the inclusion criteria implemented as compounds were randomly used to train the model. The compounds were chemically and structurally diverse. Models 2 to model 8 lost some correlations (represented by overall score). This can be explained by the random variation or statistical fluctuations [51]. However, the observed decrease in correlation does not necessarily indicate a significant decline in model performance but rather reflect the inherent variability in the data.

Fig 4A represents that scatter plot of model 1 built with KPLS_linear_36 (Table 1), which is a QSAR model generated by KPLS with linear fingerprints, had an overall score of 0,75, SD of 0,37, $R^2$ of 0,77, RMSE of 0,36 and $Q^2$ of 0,67. In general, model 1 had good predictive activity for compounds with -5 to -7 kcal/mol suggesting that for the trend line to be more linear towards the more active compounds (activity score -8 to -10 kcal/mol), more compounds need to be included in the high activity range. It is can also be noted that model1 was trained with a set of compounds with similar chemical characteristics as they are clustered around the same region.

The protocol used for building model 1 was repeated to train the second model, with inclusion of top scoring compounds, 1000 enumerated design ideas and 1000 ChEMBL database compounds for model training. The same procedure was repeated until the model reached convergence, thus improving the quality of the data by coupling high affinities PfHsp90 with activities of the compounds. As the procedure was repeated, a growing number of compounds were added to the high activity region (between -8 kcal/mol and -10 kcal/mol in model 13) (Fig 4B). Some noise displayed the Auto-QSAR plot in model 2, therefore the activity was

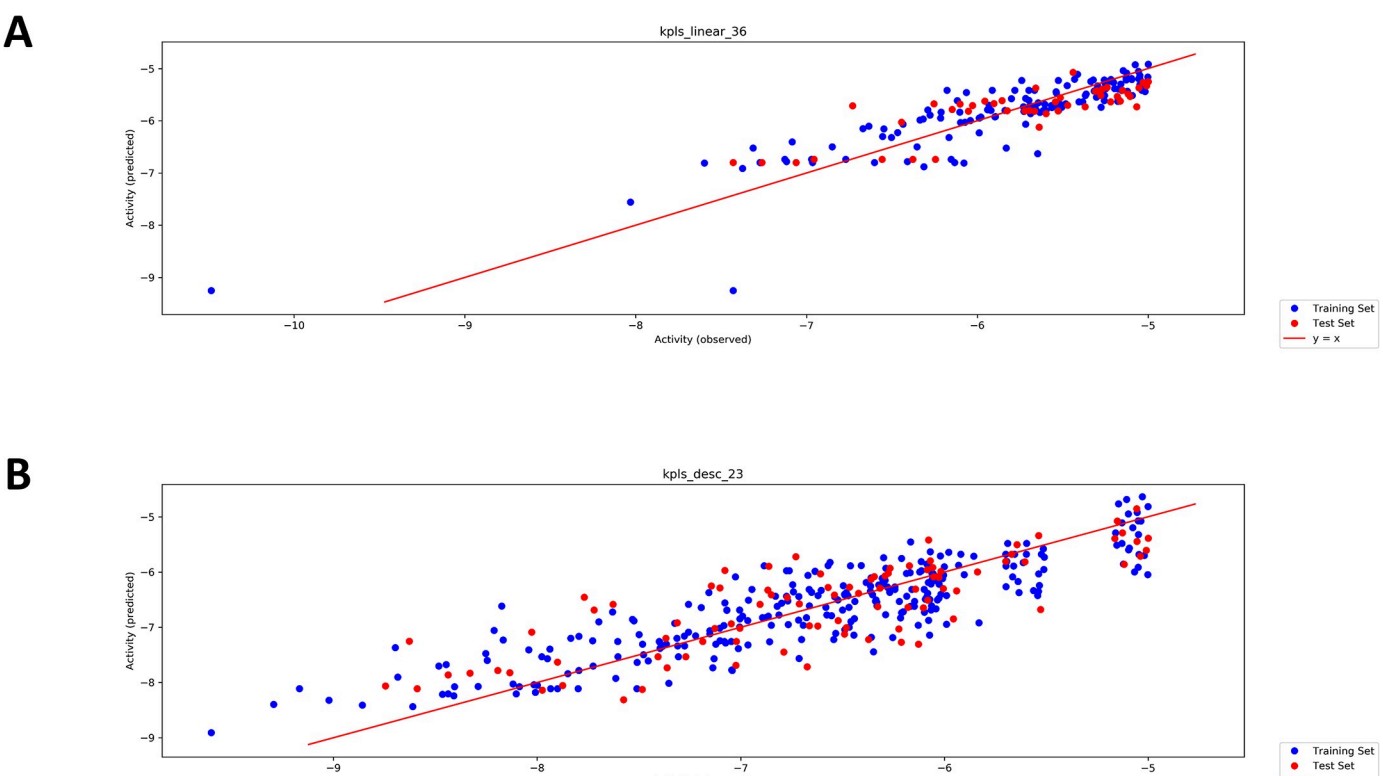

**Fig 4. Auto QSAR-AL scatter plots. A:** model 1 showing the performance of the QSAR KPLS model's predicting activity for experimental binding affinity for the test set. Model trained 123 ligands with docking scores ranging from -5.0 kcal/mol to -7.6 kcal/mol. **B:** Model 13, where the model has reached convergance. AutoQSAR randomly divided the selected ligands into the training at 75% (Blue dots) and test sets at 25% (Red dots).

adjusted to -6 kcal/mol for training models 3 to 9 (S4-S6 Figs in S1 File).To further avoid noise in the plots, models 10 and 11 (S6 Fig in S1 File) were developed specifically with ligands demonstrating docking scores ≥6.5 kcal/mol, lastly, models 12 and 13 were trained using ligands possessing docking scores of ≥-7 kcal/mol. Notably, the trend line was linear towards the observed high activity region. Model 13 (Fig 4B), which is where the model attained its convergence, denotes that the model was eventually able to recognize compounds it has not initially selected (S2-S6 Figs in S1 File).

## Induced fit docking

The AL model enhanced by classical GlideSP ligand docking yielded 236 best docked compounds, with the ability to bind to the receptor PfHsp90-NTD at the ATP binding site. The ligands exhibited docking scores of -10 kcal/mol to -6 kcal/mol, from a total of 43097 docked poses. The 236 compounds were then subjected to IFD against receptor PfHsp90. Rescoring by extra precision (XP), IFD resulted in 113 top-scoring compounds which were selected based on the docking score, e-model score, IFD score and a visual inspection of the binding mode (data not shown). It was interesting to note that, even though AutoQSAR models were only trained on compounds with docking scores between -5 and -10 kcal/mol, IFD, a more robust scoring function, was able to calculate compounds with docking scores between > -5 kcal/mol and > -13 kcal/mol. This is because IFD tends to use a more extensive sampling of ligand conformations and protein-ligand interactions compared to GlideSP resulting in increased

conformational sampling and improved accuracy of predicting ligand binding modes and binding affinities. Lipinski's rule of 5 was used to filter that compounds further to 62 compounds (docking score of -10 to -15 kcal/mol, see S2 Table in S1 File). However, most of the compounds were not commercially available and a total of 15 compounds were purchased (see S2 Table in S1 File). The IFD and MM/GBSA results of the top-scoring compounds are presented in Table 2 and the data for the reminder of the compounds can found in the S2 Table in S1 File.

Most top scoring study compounds had a higher affinity (docking score range: -10.86 to -13.49kcal/mol and ΔGBind range: -25.08 to -67.31kcal/mol) for PfHsp90 compared to the reference compound **10** (Comp 10) exhibiting docking and ΔGBind scores of -10.49kcal/mol and -28.47kcal/mol, respectively. Interestingly, the study compounds also exhibited higher affinities compared to harmine (Table 2), an established selective inhibitor of PfHsp90. It was of interest to this study to obtain compounds which binds preferentially or selectively towards PfHsp90 over human Hsp90. Therefore, human Hsp90 was also as a receptor for IFD analysis using all 62 study compounds. While high docking scores were obtained for human Hsp90 (S3 Table in S1 File range: -8.06 to -12.1kcal/mol), the binding affinities of the compounds were higher for PfHsp90 (Table 2; S2 Table in S1 File). As previously mentioned, interactions with Arg98 contribute to selectivity. When comparing the 2D interaction diagrams of PfHsp90 (S7 Fig in S1 File) with HsHsp90 (S8 Fig in S1 File), it is interesting to note that despite the disparities in the docking scores, the study compounds do not display any interaction with the selectivity conferring residue Arg98, except for FTN-T3. This may suggest that other interactions play a role in the higher docking scores observed in PfHsp90.

Visual inspection of the 2D interaction diagrams of the top-scoring compounds revealed that these are well accommodated within the ATP binding site of the PfHsp90 (Fig 5; S7 Fig in S1 File). All the compounds represented in Fig 5 display water mediated-/hydrogen bond interactions with ATP binding residues such as Asn37, Asp79, Gly83, Asn92, Lys44, Phe124, and Ala38 [33]. The binding of the compounds with these residues suggest that they will likely compete with ATP for binding PfHsp90. Interestingly, only compound FTN-T3 seems to be interacting with the selectivity conferring Arg98 via water-mediated hydrogen bond (Fig 5). Harmine, a proven inhibitor of PfHsp90, can be observed to interact with Asn37 through a water mediated hydrogen bond, as well as a salt bridge and hydrogen bond with Asp79 (Fig 5). We noted that most of the top-scoring compounds make contacts with previously described ATP binding residues, they do not interact with Arg98, which has been described to contribute to selectivity [20].

**Table 2. Induced fit docking and MMGBSA results of the top-scoring compounds.**

| Compound ID | Docking | Glide emodel | IFDScore | ΔGBind | ΔGBind Coulomb | ΔGBind Covalent | ΔGBind Hbond | GBind Lipo | ΔGBind Packing | ΔGBind Solv GB | ΔGBind vdW |
|---|---|---|---|---|---|---|---|---|---|---|---|
| **FTN-T1** | -13.49 | -83.95 | -516.39 | -67.31 | -40.70 | 2.48 | -6.28 | -6.80 | -1.67 | 25.98 | -40.31 |
| **FTN-T2** | -13.10 | -66.56 | -517.56 | -58.66 | -40.74 | 2.88 | -6.32 | -5.05 | -1.39 | 27.27 | -35.31 |
| **FTN-T3** | -10.457 | -101.9 | -523.50 | -47.95 | -37.40 | -5.47 | -6.80 | -5.59 | -1.54 | 33.26 | -35.31 |
| **FTN-T4** | -11.81 | -89.54 | -521.42 | -41.60 | -35.74 | 5.60 | -5.37 | -5.99 | -1.40 | 25.52 | -24.21 |
| **FTN-T6** | -12.05 | -63.46 | -519.19 | -35.61 | -25.42 | 3.75 | -3.73 | -6.63 | -1.63 | 29.80 | -31.76 |
| **FTN-T5** | -11.75 | -70.28 | -514.41 | -34.72 | -12.96 | 1.94 | -3.33 | -8.04 | -1.44 | 19.60 | -30.48 |
| **FTN-T9** | -10.86 | -60.37 | -518.39 | -25.08 | -31.65 | 6.60 | -3.25 | -8.47 | -3.02 | 38.98 | -24.27 |
| **Comp 10** | -10.49 | -76.16 | -532.21 | -28.47 | -16.36 | 4.24 | -3.67 | -7.57 | -4.04 | 33.01 | -34.08 |
| **Harmine** | -8.29 | -41.86 | -517.44 | -27.52 | -3.23 | 1.65 | -1.65 | -11.2 | -2.70 | 18.39 | -28.76 |

The Auto QSAR model generated compounds which were analogous to each other, compounds FTN-T1, FTN-T2, FTN-T3, FTN-T4, FTN-T6 and FTN-T9 (Table 2; Fig 5) contain a 7H-Purine scaffold, like ADP/ATP. Purine containing compounds, such as ATP and its analogs, are known to have a high affinity to the ATP-binding pocket Hsp90. This is because the purine ring system allows for the formation of multiple hydrogen bonds and hydrophobic interactions with the binding pocket residues, resulting in a higher affinity towards the protein. Compound FTN-T5 contains a 1-methylpyrimidin-2(1*H*)-one moiety, which likely played a role in its overall lower binding affinity towards PfHsp90. Compound FTN-T5 seems to be interacting to PfHsp90 through mainly hydrophobic interactions. It is possible that its overall lower docking can be attributed to the bad water contacts (Fig 5; S7 Fig in S1 File). It should be noted that the main drawback of docking is that it does not account for the effects of waters in binding affinity estimations and ligand strain energy caused by the binding event.

## Molecular dynamic simulations

The dynamic behaviour and conformational changes induced on PfHsp90 by top-scoring compounds was evaluated by molecular dynamics simulations. Fig 6(A)–6(C) show the Root Mean Square Deviation (RMSD) and Root mean fluctuations (RMSF) of the PfHsp90 and five top scoring compounds and a known inhibitor harmine. RMSDs are used to assess the average displacement between a group of atoms in a specific frame relative to a reference frame. The protein RMSD plots (Fig 6A) offers insights into the structural conformation of the PfHsp90 protein throughout the simulations. By aligning all frames on the reference backbone and calculating the RMSD based on atom selection, the stability of the protein's and the ligand structures can be effectively monitored [52]. A well-equilibrated simulation is characterized by RMSD fluctuations around 1–3 Å range for small, globular proteins. In our study, PfHsp90 remains stable in its interactions with all tested ligands, exhibiting an average RMSD of 2 Å for all complexes (Fig 6A). More considerable RMSD changes would indicate significant conformational fluctuations, with RMSD values fixed around 1.7 Å, it is deduced that PfHsp90 was greatly stable.

The ligand RMSD plots for compounds FTN-T1, FTN-T3, and FTN-T4 exhibited remarkable stability throughout the 100ns simulation, converging at average RMSD values of 1.8Å (Fig 6B). The comp**10**, FTN-T2, FTN-T5, FTN-T9 and harmine complexes, on the other hand, displayed signs of binding instability as the ligand RMSD values fluctuated throughout the simulation trajectory (Fig 6B), with compounds FTN-T2 and FTN-T5 generally equilibrating at average RMSD of 4.65 Å and 6.6 Å, respectively. The most significant deviations were seen for comp**10**, harmine and FTN-T9, however, displaying average RMSDs of 7 Å.

The RMSF plots (Fig 6C) for all simulated protein-ligand complexes provided valuable insights into the dynamic behaviour of the protein residues throughout the simulations. We observed prominent RMSF fluctuations in the residue index window spanning positions 20 to 30, 50 to 70 and 100 to 125. These regions mainly correspond to loops in PfHsp90 indicating that the loops are less structured and more flexible, leading to pronounced fluctuations during the simulation.

The 2D ligand-protein contacts diagram revealed that most of the top scoring compounds bound in the ATP binding pocket of PfHsp90. FTN-T4 displays the most stability, remaining bound throughout the 100ns simulation time. The exceptional stability of compound FTN-T1 and FTN-T4 is most likely explained by the direct strong hydrogen bond with the NH₂ group with Asp79 remaining stable for 99% and 92% of the simulation window, respectively. Phe124 maintained hydrogen bonds with the OH groups of FTN-T1 and FTN-T4, which remained stable for 84% and 60% of the simulation time, respectively. It should also be noted that the

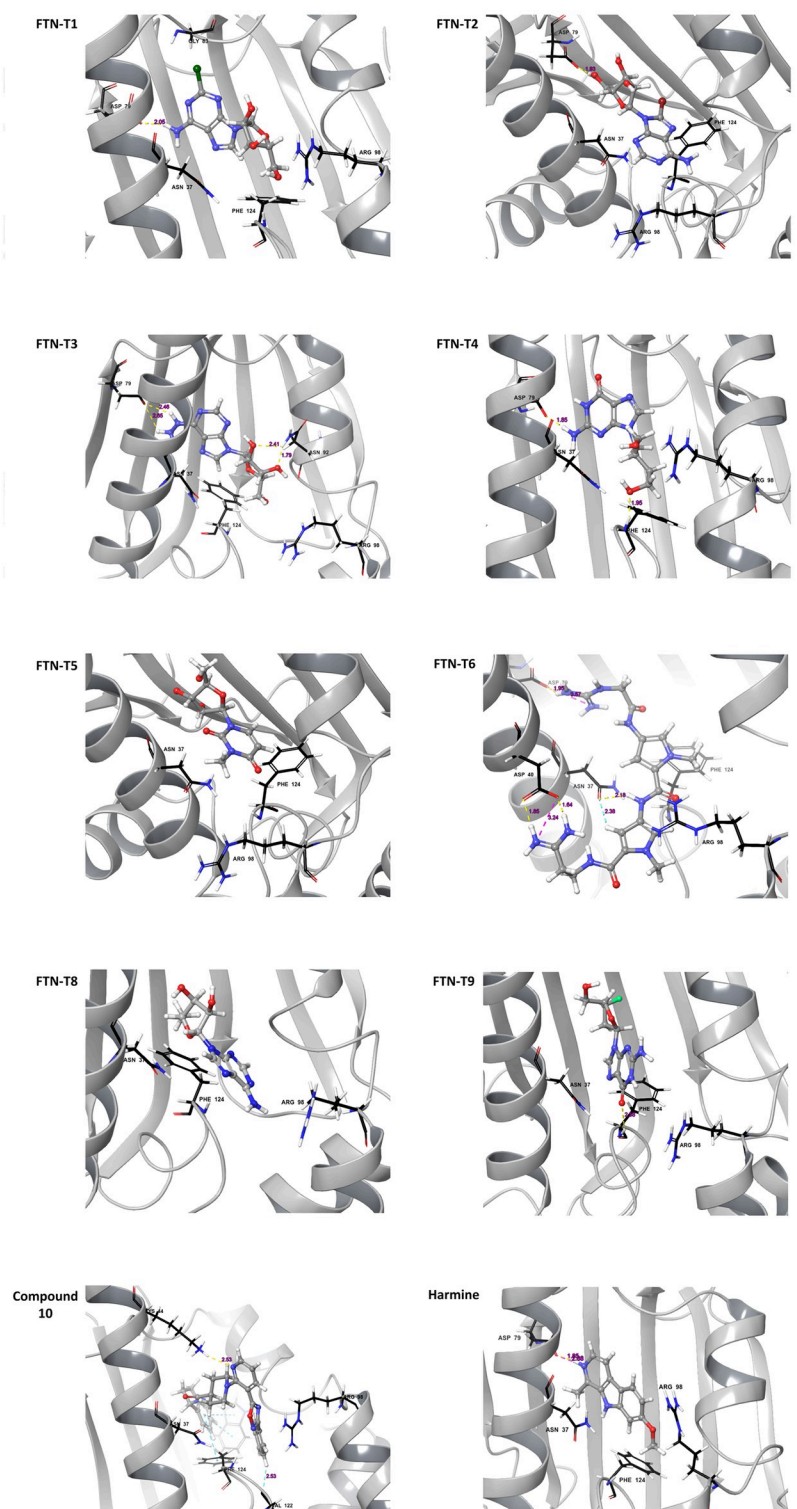

**Fig 5. 3D representations of the compounds FTN-T1, FTN-T2, FTN-T3, FTN-T4, FTN-T5, FTN-T6, FTN-T9 and Harmine in the PfHsp90-NTD binding pocket.** The 3D structure of PfHsp90 is rendered in green ribbons, with residues found at the ATP binding pocket shown in red sticks. Residues Ala38, Arg98 and Ile173 which are unique to PfHsp90 represented by blue sticks. Water molecules and hydrogen bonds represented in red and yellow, respectively.

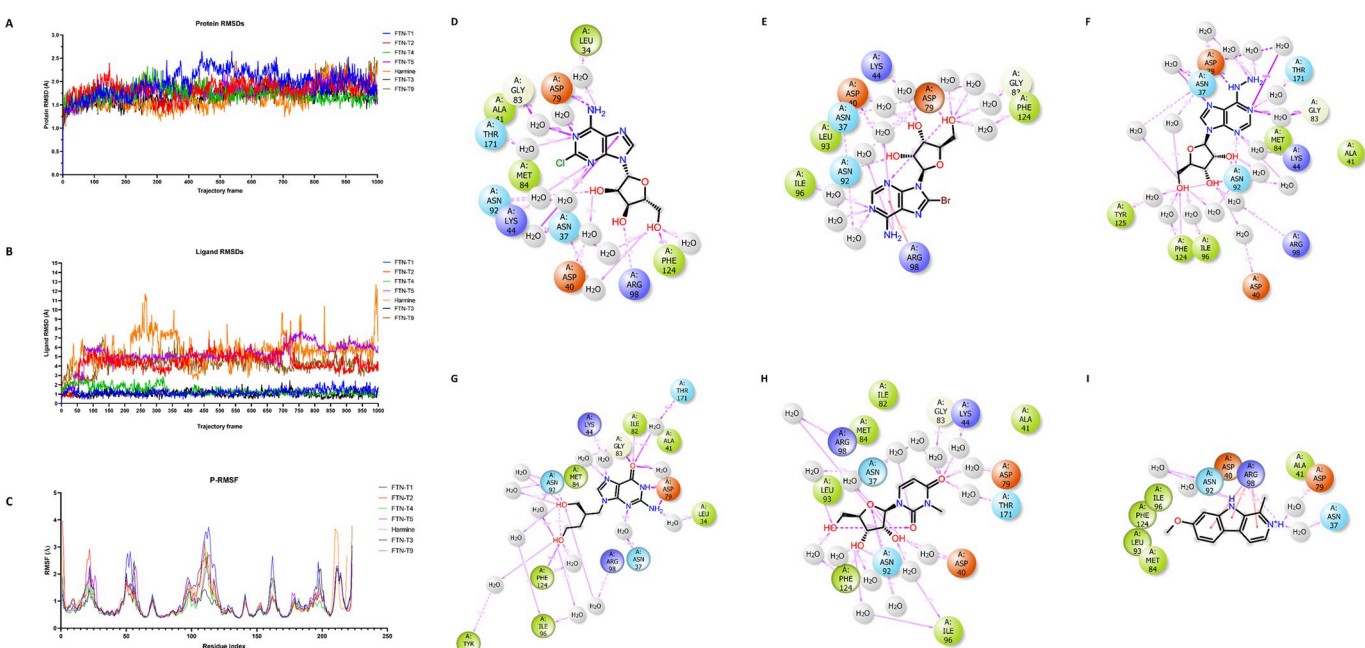

**Fig 6. Molecular dynamics simulation results for compounds FTN-T1, FTN-T2, FTN-T3, FTN-T4, FTN-T5, FTN-T9 and Harmine. A:** Ligand RMSD. **B:** Protein RMSD plots. **C:** RMSF plots. **D-I**: 2D representation of the interaction diagrams for contacts made by the compounds FTN-T1, FTN-T2, FTN-T3, FTN-T4, FTN-T5 and Harmine, in the binding pocket of PfHsp90-NTD. Reference compound represented as COMP 10.

simulation interaction diagram (SID) from MDS ([Fig 7]) managed to capture some important hydrogen bond interaction network involving water. Compound FTN-T1 makes a direct contact with Arg98. Meanwhile compounds FTN-T3, FTN-T4 and FTN-T5 form water-mediated hydrogen bonds. Similar to harmine which interacts with Arg98 via three Pi-cation interaction, compound FTN-T2 similarly interacts with Arg98. The data from the 2D interaction diagrams of the study compounds and harmine inspired the prediction that these compounds would likely be selective. Even though MDS does'nt measure the effects of favourable and unfavourable waters as a docking post-processing approach. Methods such as MM-GB/SA [47] and more recently Water Map [53, 54], IFD-MD [55] prior to FEP+ were introduced to account or this challenge. In future studies we will further explore the great utility of the later approaches in our lead optimization efforts.

## Anti-*Plasmodium* and cytotoxicity of promising compounds

The 15 compounds were tested *in vitro* for anti-plasmodial activity and profiled for cytotoxicity against the CHO and HepG2 cell lines. Most of the compounds were inactive (PfNF54-IC$_{50}$ $\geq$ 6 µM) (data not shown) and at least four compounds showed moderate activity and reasonable selectivity indeces of $\geq$9, ([Table 3]), suggesting a good safety profile except for compound FTN_T2. Compound FTN-T5 showed promising anti-plasmodial activity (PfNF54-IC$_{50}$ < 1.5 µM) and low cytotoxicity was observed in CHO and HepG2 cell lines as displayed by good selectivity margins (average SI $\geq$ 30) ([Table 3]). The anti-*Plasmodium* obtained for all the study compounds were lower than harmine ([Table 3]) and the known inhibitor geldanamycin (IC$_{50}$ = 0.02µM; (11)). However, compound FTN-T5 displayed activity comparable to the reference compound **10** ([Table 3]).

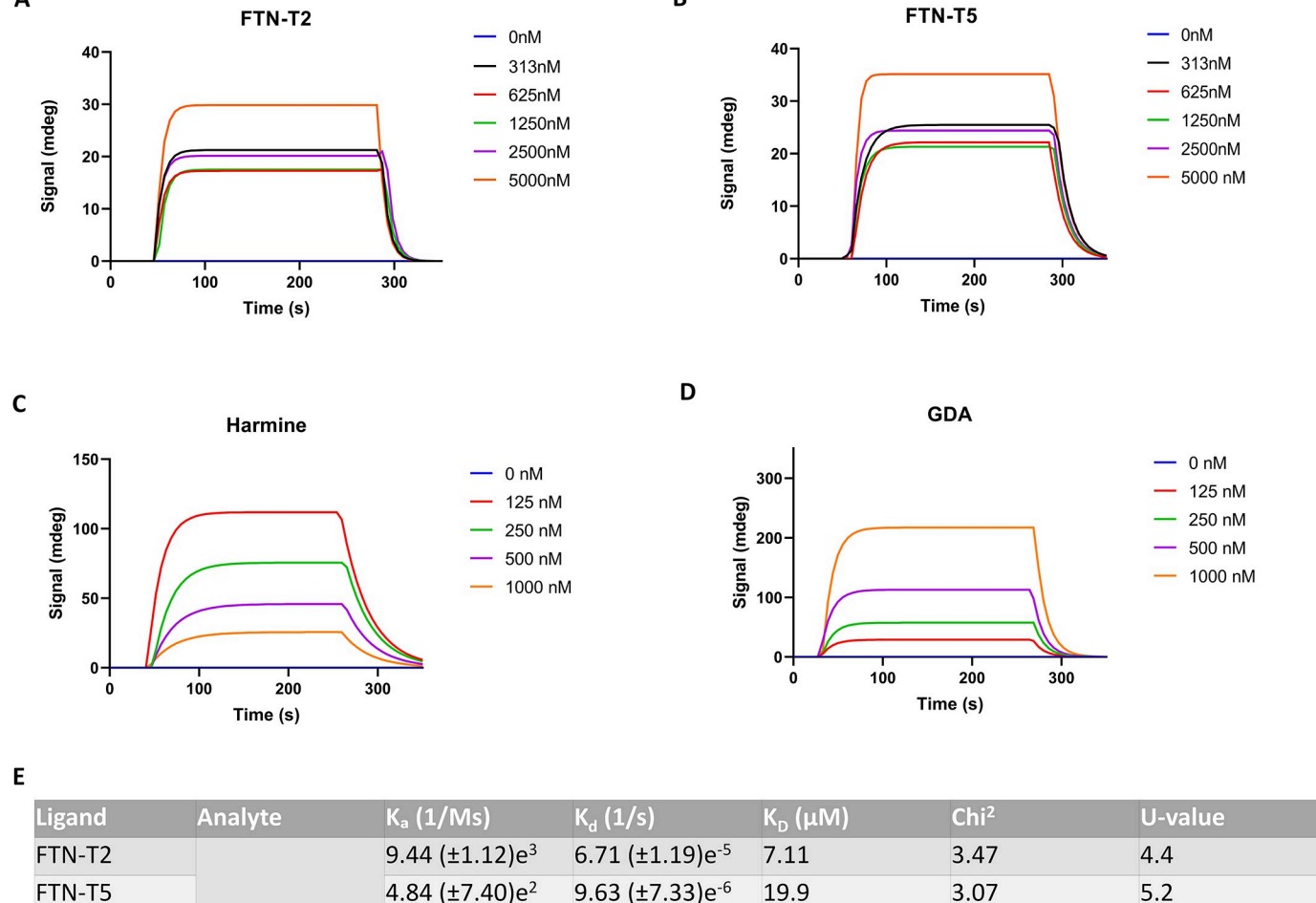

**Fig 7. SPR sensorgrams for compounds FTN-T2, FTN-T5, harmine and GDA and Kinetics constants measured by SPR for the interaction between tested compounds and immobilized PfHsp90 (E).** The association rate constant is represented by $K_a$ (1/Ms), dissociation rate constant $K_d$ (1/s), and the equilibrium constant denoting affinity $K_D$ (μM).

**Table 3. Table displaying IC$_{50}$ values in μM of promising compound activity against *P. falciparum* cells, as well as cytotoxicity towards human cells and respective selectivity indices.**

| COMPOUND ID | *In vitro* IC$_{50}$ (μM) | | | |
|---|---|---|---|---|
| | *Pf*NF54/ *Pf*3D7 *Pf*3D7*Pf*3D7 | Mammalian cells (SI) | HepG2(SI) | Study |
| FTN-T2 | 4.49 ± 1.51 | >50 (>8) | 1.50 (0) | This study |
| FTN-T3 | 5.42 ± 0.58 | >50 (>9) | >50 (>9) | This study |
| FTN-T4 | 5.01 ± 0.99 | >50 (>10) | 27.71(6) | This study |
| FTN-T5 | 1.44 ± 0.50 | >50 (>35) | 42.29(29) | This study |
| FTN-T9 | 4.86 ± 1.14 | >50 (>10) | >50 (>10) | This study |
| Harmine | 0.05 ± 0.00 | ND | ND | Shahinas *et al.*, 2010 |
| Compound 10 | 1.11 ± 0.97 | >24 (>21) | >24 (>21) | Everson *et al.*, 2021 |

*Mammalian cells refer to either Chinese hamster ovarian cells (CHO) or human fibroblast cell line.

## Binding affinities of promising compounds

Surface plasmon resonance is a technique that reveals information regarding binding affinities, and the kinetics parameters representing the interaction of a protein and its ligands. Fig 7 displays the SPR sensorgram for the interactions between compounds FTN-T2 and FTN-T5 and PfHsp90-NTD (Fig 7A and 7B) aswell as harmine (Fig 7C) and Geldanamycin (Fig 7D). The sensorgrams exhibit a dose-dependent response of these compounds when interacting with the immobilized PfHsp90-NTD. FTN-T2 displays a modest binding affinity ($K_D$ = 7 μM; Fig 7E) comparable to GDA but weaker than harmine. FTN-T5, on the other hand, displays a $K_D$ of 19 μM, which falls within the same order of magnitude as the reported affinity of ADP [19] for PfHsp90-NTD. Altogether, FTN-T2 shows promising affinity for PfHsp90-NTD while the affinity of FTN-T5 show somewhat weak affinity.

## Structure-activity relationship analysis

As mentioned previously, compound 10 was utilized to generate analogues using Pathfinder reaction-based enumeration and AutoQSAR. The identified top scoring compounds were subjected to various *in vitro* and *in silico* experiments to determine their respective activities. According to *in silico* data, the generated compounds display improved binding energies, compared to compound 10, when bound to PfHsp90. The compounds FTN-T1, FTN-T2, FTN-T3, FTN-T4, FTN-T8 and FTN-T9 contain purine moieties, while FTN-T5 consists of a pyrimidine. Purine-based compounds possess the ability to form multiple hydrogen bonds as well as hydrophobic interactions that result in higher bonding affinity. Purine groups containing halogens exhibit the highest docking scores, as observed from the docking scores of FTN-T1 (-13.49 kcal/mol) and FTN-T2 (-13.10 kcal/mol). However, FTN-T1 (consisting of a chloride bonded to the carbon at the 2-position of the purine) seems to be stable in the binding pocket of PfHsp90, as evidence by ligand RMSD values that equilibrate around 1.8Å (Fig 6B), as opposed to FTN-T2 (consisting of a bromide bonded to the carbon at the 8-position of the purine) at around 4.65Å after 100ns. Compared to FTN-T1, FTN-T2 and FTN-T3, which consist of a purine moiety, FTN-T5 consists of a pyrimidine.

It is worth noting that despite the lower docking score of FTN-T5, it displayed the highest antiplasmodial activity among the purchased compounds (Table 3; $IC_{50}$ = 1.44 μM), followed by FTN-T2 (Table 3; $IC_{50}$ = 4.49 μM). On the other hand, corroborating the docking results, the purine moiety containing FTN-T2 (Fig 7; $K_D$ = 7.11 μM) displays a higher binding affinity for PfHsp90, compared to the pyrimidine containing FTN-T5 (Fig 7; $K_D$ = 19 μM), as observed from SPR results.

## Discussions

Malaria remains a global public health concern, necessitating the discovery of novel therapeutics to combat drug-resistant strains [56]. The PfHsp90 protein, a molecular chaperone, is critical in the parasite's survival and virulence [7, 8]. Therefore, by targeting PfHsp90, we set out to generate starting points for effective antimalarial drugs, likely to circumvent drug resistance. In this study, we employed an innovative approach, Auto-QSAR-based Active Learning Docking, to generate potential inhibitors of PfHsp90 as promising anti-*Plasmodium* agents. Reaction-based enumeration was implemented to convert compound 10 to generate 10 000 design ideas. While ultra-large screening of large libraries of compounds has previously been achieved [57], the number of ligands produced would have been computationally expensive for our resources. We, therefore, devised an AL enhance by docking protocol to overcome the resource limitation while transversing a large chemical space in a relatively short time. The protocol was employed to enrich the data set before subjecting

them to IFD. A is a classification under supervised machine learning techniques that develops highly accurate models, effective for exploring chemical space with docking and deep learning as a substitute for complex or expensive implement scoring functions [58]. The previous studies of [43, 57] have applied active learning in the drug discovery field and were able to demonstrate that ligand-based QSAR models are capable of "learning" a docking score over a particular domain, in applications including molecular docking and free energy calculations, while significantly lowering the computational expenses of screening an extensive library.

The capabilities of the AL approach coupled with docking to screen an extensive database were demonstrated by the 13 rounds of screening the design ideas with Glide SP docking and predicting their docking scores using various AutoQSAR models prior to docking compounds. The active learning model was able to "learn" how to predict the activity of highly active compounds against PfHsp90, from the initial point where the model could only predict with the highest level of accuracy compounds in the ranges of -5 kcal/mol to -7 kcal/mol as the model was only trained in this set of compounds and having similar chemical characteristics. With the addition of more compounds with different chemical properties to the model, more compounds can be seen in the high activity region ranging between -8 kcal/mol and -10 kcal/mol (S2 to S6 Figs in S1 File), suggesting that the model was able to learn how to predict the activity of highly active compounds. The active learning model yielded 236 ligands promoted to the following filtering point: IFD and Molecular dynamics simulations. The strong complementarity affinities of these compounds for PfHsp90 could be explained by due to their resemblance of its natural ligand. Thus, it is most likely it is proposed that these ligands might outcompete ADP/ATP given their high docking scores.

The 15 purchased compounds were evaluated for anti-*Plasmodium* activity. The findings highlight the promise of FTN-T5 as a hit compound for further optimization, given its promising activity against the PfNF54 strain of P falciparum. FTN-T5 manifested low cytotoxicity towards CHO and HepG2 cell lines. Biophysical investigations revealed that FTN-T5 binding to PfHsp90-NTD with weak affinity. The binding affinity data was not surprising as MDS predicted high fluctuations in the PfHsp90-FTN-T5 complex, suggesting a degree of non-specific binding. While it is possible that the weak affinity of FTN-T5 could be explained by pan-inhibition of other Hsp90 isoform due to the similarity of the ATP binding pocket architecture, the observed discrepancy between the anti-*Plasmodium* activity and binding affinity of FTN-T5 warrants further investigations. We suggest methods such as pull down assays as FTN-T5 displayed low cytotoxicity, raising the prospect that it does not bind to human Hsp90.

FTN-T5, a pyrimidine-based compound, displaying fewer interactions with the ATP binding pocket residues. It appears that pyrimidine-based compounds may have reduced affinity for Hsp90 compared to purine-based compounds. Compound FTN-T2 seems to bind to PfHsp90 with modest affinity, comparable to GDA. FTN-T2 contain a purine moiety, exhibit strong interactions with the ATP-binding pocket of PfHsp90. The purine ring system facilitates the formation of multiple hydrogen bonds and hydrophobic interactions with key binding pocket residues, leading to higher affinity for the protein.

Overall, this study identity compound FTN-T5 and FTN-T2 as promising starting point for future multiparameter optimization to improve their binding affinity and potency. Given that PfHsp90 is a promising drug target in malaria, we believe that the study has contributed more compounds to be explored for optimization efforts. As the search for effective anti-malarial agents continues, these findings advocate for the iterative refinement of FTN-T2 and FTN-T5, using its favorable attributes while addressing some of their weakness.

### Limitations of the study

The MMGBSA and and induced fit docking which consider target site to be flexible add too much variability for the used machine learning model.

## Supporting information

**S1 File.**
(DOCX)

## Acknowledgments

We thank the Centre for high performance (CHPC), CSIR, South Africa for providing computing facilities. We are grateful to the holistic drug discovery and development (H3D) testing centre based that the University of Cape Town for screening our compounds against *P. falciparum* and perfoming cytotoxicity assays.

## Author Contributions

**Conceptualization:** Thato Matlhodi, Njabulo Joyfull Gumede, Fortunate Mokoena.

**Formal analysis:** Thato Matlhodi, Lisema Patrick Makatsela, Tendamudzimu Harmfree Dongola, Mthokozisi Blessing Cedric Simelane, Addmore Shonhai.

**Funding acquisition:** Fortunate Mokoena.

**Investigation:** Thato Matlhodi, Njabulo Joyfull Gumede, Fortunate Mokoena.

**Supervision:** Njabulo Joyfull Gumede, Fortunate Mokoena.

**Validation:** Mthokozisi Blessing Cedric Simelane, Addmore Shonhai, Njabulo Joyfull Gumede.

**Writing – original draft:** Thato Matlhodi, Fortunate Mokoena.

**Writing – review & editing:** Thato Matlhodi, Lisema Patrick Makatsela, Addmore Shonhai, Njabulo Joyfull Gumede.

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
