## [Decision Letter · Decision Letter 0]

24 Mar 2024

PONE-D-24-01107Auto QSAR-based Active learning docking for hit identification of potential inhibitors of Plasmodium falciparum Hsp90 as antimalarial agentsPLOS ONE

Dear Dr. Mokoena,

Thank you for submitting your manuscript to PLOS ONE. After careful consideration, we feel that it has merit but does not fully meet PLOS ONE’s publication criteria as it currently stands. Therefore, we invite you to submit a revised version of the manuscript that addresses the points raised during the review process.

**Academic Editor**:

There is a need to rationalize all the assumptions made in your study. The reviewers have made general and specific recommendations that should be implemented and/or responded to for clarity and better understanding by our readers.

Specifically, address the following concerns: 

1. Resolution of all images. 

2. Homology modelling between the ATP binding pockets of the Plasmodium falciparum heat shock protein 90 (PfHsp90) and the human heat shock protein 90 (hHsp90). Are there any differences in amino acids sequence that could drive selective toxicity? 

3. Pharmacophore modelling based on the study conducted. From this study, what are the probable binding modes and chemical interactions between the tested compounds and the molecular target (ATP binding pocket of the PfHsp90)? What are the chemical groups essential for the suggested chemical interactions?

4. It would be prudent to depict the interactions between the most active novel compound and the molecular target (PfHsp90). 

5. Could the Authors speculate to what extent the identified compounds could be applicable to the other plasmodial species that infect humans. 

We look forward to receiving your revised manuscript.

Kind regards,

Peter Mbugua Njogu, Ph.D.

Academic Editor

PLOS ONE

Journal Requirements:

3. Thank you for stating the following financial disclosure: "FM was awarded the Grand challenges Africa drug discovery seed grant (GCA/Round10/DD-065), funded by the Bill and Melinda Gates foundation is hereby acknowledged. "

Reviewers' comments:

Reviewer's Responses to Questions

**Comments to the Author**

1. Is the manuscript technically sound, and do the data support the conclusions?

Reviewer #1: Yes

Reviewer #2: Partly

2. Has the statistical analysis been performed appropriately and rigorously? 

Reviewer #1: Yes

Reviewer #2: Yes

3. Have the authors made all data underlying the findings in their manuscript fully available?

Reviewer #1: No

Reviewer #2: Yes

4. Is the manuscript presented in an intelligible fashion and written in standard English?

Reviewer #1: Yes

Reviewer #2: Yes

5. Review Comments to the Author

Reviewer #1: Comments to the Author

The manuscript entitled: “Auto QSAR-based Active learning docking for hit identification of potential inhibitors of Plasmodium falciparum Hsp90 as antimalarial agents” investigates the application of molecular docking and active learning (AL) models to discover novel inhibitors of PfHsp90, a validated antiplasmodial target. The authors were able to purchase 15 new compounds for validation against PfHsp90 using AL models, demonstrating affinities ranging from 13.5 – 19.9 µM and IC50 values lower than 6 µM in in vitro assays against the parasite. However, several critical issues require attention, necessitating a major revision before the acceptance of the paper. Primarily, the quality of the images provided is inadequate, impairing the analysis of results. Enhanced image quality must be done to facilitate comprehensive data interpretation. Additionally, the inclusion of the 2D structures of the purchased compounds should be added (this is very important) within the main manuscript. Also elucidating their binding modes in the active site environment of PfHsp90 would greatly benefit readers' understanding. In addition, a SAR analysis must be done for these new inhibitors, based on their affinity to the protein target (PfHsp90) and according to their potency against the parasite, highlighting the significance of specific scaffolds or substituent groups in driving observed activity. Finally, some points in the manuscript are too extensive and should be more concise to facilitate readability, in many points is difficult to understand what the authors want to express. Overall, while the findings of this study hold promise for publication in PLOS ONE, substantial revisions are required to address the aforementioned issues as well as the following points.

- All figures must be recreated with significantly higher resolution, as the current images lack clarity and hinder comprehensive analysis.

- There are instances within the manuscript, such as in line 472, where the authors have employed a comma (',') as the decimal separator in numerical values. It is recommended that they review and replace these commas with periods ('.') for consistency and clarity.

- Throughout the paper. Double-check when Plasmodium, Pf, Plasmodium falciparum, etc¸ are cited since these should be written in italics (PfHsp90 -> PfHsp90). IC50 -> IC50; chEMBL -> ChEMBL;

- The chemical structures of the novel and purchased compounds must be represented in the manuscript. As well as a detailed SAR analysis of these compounds against the protein and the parasite.

- Line 60. In the introduction, explain more about how climate conditions complicate the treatment of malaria. I don’t think the right word should be ‘complicates’ but something more related to increasing the risk of more people being infected with malaria.

- An image comparing PfHsp90 with human Hsp90 active sites and highlighting the differences and the amino acids commented on by the authors would be beneficial for the readers.

- From lines 108 to 115 I don’t think this is necessary to be said. Also, the example of using ML of A2A receptor antagonist should be replaced by some examples of using ML to design new antiplasmodial/antimalarial inhibitors. This will be better for the audience reading it and make a direct correlation between the use of ML to design new inhibitors targeting malaria.

- In line 130, the authors write ‘AL models’ but no definition about what is the abbreviation AL (Active learning) was given before.

- Line 272, why did the author select the division 75/25 for the training set?

- Line 275, what about the MAE analysis of the created model?

- In docking methodology, the authors should say that they also have made a visual inspection in addition to the docking score to analyze the obtained poses (see ref https://pubs.acs.org/doi/abs/10.1021/acs.jmedchem.0c02227). Moreover, what were the interactions analyzed to perform the visual inspection, this could be helpful to other analyzing inhibitors against PfHsp90.

- Line 410 ‘pi-pi’ -> π-π

- In the "Induced fit docking of reference compounds" section of the results, it is necessary for the authors to include an image illustrating the ligand(s) positioned within the active site, emphasizing the primary interactions discussed. While it may serve as a complement to Figure 3, it's important to note that the resolution of Figure 3 is insufficient, rendering it impossible to discern details. Therefore, a new figure with higher resolution should be provided to adequately visualize the ligand interactions in the active site.

- Line 424, “10, 000” -> 10000

- It is not usual the construction of AL models using docking scores, the authors should provide a discussion about that with recent literature in which the use of AL models created by docking results resulted in outstanding outcomes. Usually, more robust calculations are performed for AL models (e.g. FEP) – check references: https://www.sciencedirect.com/science/article/pii/S2667318522000204 and https://pubs.acs.org/doi/10.1021/acs.jcim.3c00681.

- Line 473, the correct is ‘predictive affinity’ since this is a regression model using Gibbs free energy provided by docking or experimental Ki/Kd. Activity is for a classification model (not for regression model), in which a compound is active or inactive. Check this in other parts of the manuscript as well, please.

- In Figure 4, the X and Y axis labels should be changed to Affinity instead of Activity.

- Line 478, what about the applicability domain? I’m not sure if the authors said something about this in the manuscript.

- Results sections – ‘induced fit docking’ - An image of the best novel and predicted compounds in the active site with the intermolecular interactions with the target should be provided.

- Table 2: What is the energy unit in Table 2? kcal/mol? In addition, Table 2 must be adapted to fit on the page.

- In line 531, it is imperative to elucidate the disparities in interactions that lead to variations in docking scores between compounds targeting PfHsp90 and hHsp90. Furthermore, the text should explicitly outline the key interactions necessary to confer selectivity for PfHsp90 over hHsp90. Clarification on these points is essential for a comprehensive understanding of the molecular basis underlying the selectivity of compounds toward PfHsp90.

- Figure 5 must be done again to improve the resolution and coloring of the atoms of the ligand according to the heteroatoms to facilitate the visualization. All atoms with the same color for the ligand make it hard to analyze the pose.

- Line 538, ‘these residues suggest that they will likely compete with ATP for binding PfHsp90’, experimental biochemical assays were done using PfHsp90, right? I suppose the authors have checked the compounds' inhibition mechanism against PfHsp90. In this way, the ‘likely’ word here can be replaced.

- ‘Molecular dynamic simulations’ -> Molecular dynamics simulations

- In the "Molecular dynamics simulations" section, the analysis of results was hindered by the low resolution of Figure 6, rendering it impossible to interpret. Additionally, the authors must conduct supplementary analyses on the MD simulation trajectories. For example, they should examine and plot the distances of the hydrogen bonds formed between the compounds and the protein target throughout the simulation. Furthermore, it is essential to verify the persistence of these hydrogen bonds after a 100 ns simulation to provide a more comprehensive assessment of compound stability in conjunction with RMSD analysis. Relying solely on RMSD/RMSF may not be sufficient to ensure thorough evaluation.

Reviewer #2: Matlhod et al used compound 10 as a reference PfHsp90 to generate 10000 models using combinations of computational techniques. These models, together with compounds from the CheMBL data base were subjected to several training and testing rounds to identify compounds with high binding affinity to PfHsp90. A few of the top rank compounds were purchased and evaluated in vitro against the drug susceptible strain of Plasmodium falciparum. I will recommend publication of the work subject to the following revision:

(1) The authors should screen at least the best in vitro hit compound from this study against drug resistant strain of Plasmodium falciparum.

(2) There are several repetitions in the manuscript, especially between the methods and results sections, consider cutting back on these.

(3) The manuscript should be sent for thorough language editing as there many incomplete sentences or inappropriate punctuations.

(4) In both experimental parts, authors have written “Survival was plotted”. This is not very correct, it must be rephrased.

(5) Since computational methods suggested high binding to humanHsp90, others might consider testing for toxicity against another human cell line like HEK293

Others

Page 11, line 62-65, rephrase the sentence ‘’ Furthermore, areas such as Rwanda (4) and East Asia (5) have begun to report the spread and dissemination of first-line treatment options to address artemisinin-tolerant P. falciparum strains to emphasize the urgent need to develop potent and reliable anti-parasitic drugs’’. It is ambiguous.

Page 11, line 67, reference needed at the end.

Page 12, line 75 to 77, references needed at the end of each sentence.

Page 14, line 131, a figure containing the structure of compound 10 should be included nearby and mentioned in the text.

Page 20, line 269, “≥6.5” double check that a minus sign(-) is not missing

6. PLOS authors have the option to publish the peer review history of their article (what does this mean?). If published, this will include your full peer review and any attached files.

Reviewer #1: No

Reviewer #2: No

---

## [Author Response · Author response to Decision Letter 0]

17 May 2024

Response to reviewer 1

General comments 

However, several critical issues require attention, necessitating a major revision before the acceptance of the paper. Primarily, the quality of the images provided is inadequate, impairing the analysis of results. Enhanced image quality must be done to facilitate comprehensive data interpretation.

Response: We have done our best to improve the quality of the images that we generated. We hope that this improvement reflects to the satisfaction of the editor and the reviewers. 

Additionally, the inclusion of the 2D structures of the purchased compounds should be added (this is very important) within the main manuscript. 

Response: The 2D structure of the purchased compounds have been provided in the supplementary data. 

Also elucidating their binding modes in the active site environment of PfHsp90 would greatly benefit readers' understanding. 

Response: The study aimed to generate novel inhibitors of PfHsp90 targeting the N-terminal domain of the protein. These would compete with ATP/ADP for binding based on the nature of the compounds' interactions with PfHsp90. In the case of inhibitors interacting with Arg98, we speculate that these would interact selectively. We have revised the results sections to make this clear. Please refer to the track change version to see the extent of the changes reflected. 

In addition, a SAR analysis must be done for these new inhibitors, based on their affinity to the protein target (PfHsp90) and according to their potency against the parasite, highlighting the significance of specific scaffolds or substituent groups in driving observed activity. 

Response: Thank you. Pleaser refer to page 28 of the manuscript line 611-633 . 

Finally, some points in the manuscript are too extensive and should be more concise to facilitate readability, in many points is difficult to understand what the authors want to express.

Overall, while the findings of this study hold promise for publication in PLOS ONE, substantial revisions are required to address the aforementioned issues as well as the following points.

Response: We have done our utmost best to revise and remove redundancies. Please refer to the track changed version to assess the extend of our effort. We hope this has made things less extensive. 

Specific comments 

All figures must be recreated with significantly higher resolution, as the current images lack clarity and hinder comprehensive analysis.

Response: Thank you for your constructive feedback regarding the figures in our manuscript. We acknowledge the importance of clear and high-resolution images for comprehensive analysis and have recreated all figures at a significantly higher resolution.

- There are instances within the manuscript, such as in line 472, where the authors have employed a comma (',') as the decimal separator in numerical values. It is recommended that they review and replace these commas with periods ('.') for consistency and clarity.

Response: Table 1 has been revised to have period to separate the decimals. Thank you for this observation. Please see revision on

- Throughout the paper. Double-check when Plasmodium, Pf, Plasmodium falciparum, etc¸ are cited since these should be written in italics (PfHsp90 -> PfHsp90). IC50 -> IC50; chEMBL -> ChEMBL;

Response: The editorial changes have been effected through the manuscript. 

- The chemical structures of the novel and purchased compounds must be represented in the manuscript. As well as a detailed SAR analysis of these compounds against the protein and the parasite.

Response: Thank you for the suggestion. The reference compound 10, harmine, as well as the top scoring compounds (purchased) were compiled in one image. Additionally, a SAR section of these compounds has been included starting from line 661-633.

- Line 60. In the introduction, explain more about how climate conditions complicate the treatment of malaria. I don’t think the right word should be ‘complicates’ but something more related to increasing the risk of more people being infected with malaria.

Response: Thank you. This has been revised to “It has also been suggested that changes in climate conditions such as increased temperatures and heavy rainfall may result in an increased mosquito population, putting more people at risk of contracting malaria (1).” Please see line 70-73. 

.

- An image comparing PfHsp90 with human Hsp90 active sites and highlighting the differences and the amino acids commented on by the authors would be beneficial for the readers. 

Response: Thank you for your feedback. The image displaying both ATP binding pockets of PfHsp90 and HsHsp90 has been provided. This image also displays the residues involved in inhibitor binding, as well as the glycine rich loop (GHL).

- From lines 108 to 115 I don’t think this is necessary to be said. Also, the example of using ML of A2A receptor antagonist should be replaced by some examples of using ML to design new antiplasmodial/antimalarial inhibitors. This will be better for the audience reading it and make a direct correlation between the use of ML to design new inhibitors targeting malaria.

Response: Thank you for your insightful feedback. We have removed the mentioned lines (108-115) as per your suggestion. Moreover, we appreciate your recommendation to replace the example with instances of using ML to design antiplasmodial/antimalarial inhibitors. We want to emphasize that this study marks the pioneering use of ML in designing such antimalarial inhibitors. 

- In line 130, the authors write ‘AL models’ but no definition about what is the abbreviation AL (Active learning) was given before.

Response: We beg the pardon of the reviewer. The definition was provided. We have now put ML/AL as abbreviations of machine learning and active learning in brackets to clarify things better. 

- Line 272, why did the author select the division 75/25 for the training set?

Response: The 75/25 training and testing data split, commonly used in machine learning, was chosen deliberately to ensure model training on the majority of data while evaluating its performance on unseen data. Initially, we aimed to ensure a diverse representation of compounds across the entire spectrum of docking scores, ranging from -6 kcal/mol to the highest docking scores. This approach is consistent with recommendations in the literature to encompass a wide range of chemical space to enhance model robustness and generalization (Gumede, 2022).

Furthermore, during the initial stages of model training, we observed that the majority of compounds were clustered within the lower docking score range. To address this imbalance and promote learning across the entire spectrum of docking scores, we opted for a 75/25 division. This decision was aimed at broadening the coverage of the model and enabling it to effectively predict the activity of compounds with higher docking scores.

- Line 275, what about the MAE analysis of the created model?

Response: Thank you for your query regarding the Mean Absolute Error (MAE) analysis of our created model. The MAE analysis was conducted and the results of our model indeed range from 1 to -1 (Table S5), indicating the accuracy of our predictions. As MAE measures the average absolute difference between predicted activities and observed activities, a value closer to 0 signifies greater accuracy in the model's predictions. 

- In docking methodology, the authors should say that they also have made a visual inspection in addition to the docking score to analyze the obtained poses (see ref https://pubs.acs.org/doi/abs/10.1021/acs.jmedchem.0c02227). Moreover, what were the interactions analyzed to perform the visual inspection, this could be helpful to other analyzing inhibitors against PfHsp90.

Response: Thank you for the feedback. The implementation of visual inspection to assess the binding interactions between the ligands and protein, in combination with docking score, IFD score and e-model score, was stated. We do acknowledge the need to elaborate on the nature of the binding interactions that need to be observed to assert selectivity towards PfHsp90, and thank the reviewer for taking note of this. This addition was made in the induced fit docking methodology.

- Line 410 ‘pi-pi’ -> π-π

Response: Thank you for bringing to our attention the need for the correct representation of 'pi-pi' interactions as 'π-π' in line 410 of the manuscript. We have promptly made the necessary correction to ensure accuracy in our terminology.

- In the "Induced fit docking of reference compounds" section of the results, it is necessary for the authors to include an image illustrating the ligand(s) positioned within the active site, emphasizing the primary interactions discussed. While it may serve as a complement to Figure 3, it's important to note that the resolution of Figure 3 is insufficient, rendering it impossible to discern details. Therefore, a new figure with higher resolution should be provided to adequately visualize the ligand interactions in the active site.

Response: Thank you for your valuable feedback regarding the "Induced fit docking of reference compounds" section in our results. We recognize the importance of providing a clear visualization of ligand interactions within the active site. To address this concern, we have included a new figure with higher resolution, emphasizing the primary interactions discussed. This figure serves as a complement to Figure 3, which we acknowledge had insufficient resolution for discerning details. We trust that the inclusion of this new figure enhances the clarity and comprehensibility of our findings

- Line 424, “10, 000” -> 10000

Response: Thank you for pointing out the need to correct the formatting of "10,000" to "10000" in line 424 of the manuscript. We have made the necessary adjustment to ensure consistency in formatting throughout the document

- It is not usual the construction of AL models using docking scores, the authors should provide a discussion about that with recent literature in which the use of AL models created by docking results resulted in outstanding outcomes. Usually, more robust calculations are performed for AL models (e.g. FEP) – check references: https://www.sciencedirect.com/science/article/pii/S2667318522000204 and https://pubs.acs.org/doi/10.1021/acs.jcim.3c00681.

Response: Thank you for your insightful comments regarding the construction of AL models using docking scores. We acknowledge that this approach may seem unconventional, as docking scores are typically not the sole criterion for model training in traditional machine learning applications in drug discovery. However, our methodology aligns with recent innovations where docking scores are integrated into AL frameworks to refine and prioritize the selection process for subsequent rounds of simulation or experimental validation. In response to your comment, we have included a more detailed discussion on the use of docking scores in AL models (line 129 – 143). 

“Recent studies demonstrate the efficacy of Active Learning (AL) models that incorporate docking scores in streamlining the drug discovery process. Notably, Marin et al.(2024) study on regression-based AL models highlights how these can rapidly prioritize compounds in large-scale docking, enhancing efficiency and reducing costs. Similarly, Aniceto et al. (2023) applied AL to optimize virtual screening processes for urease inhibitors, demonstrating the model's ability to increase hit rates effectively.

Another significant contribution comes from Gumede (2022) study, which combines QSAR-based AL with docking scores to prioritize SARS-CoV-2 PLpro inhibitors. This innovative approach has shown that AL models can effectively select promising candidates for further computational and experimental analysis, accelerating the development of therapeutic agents

These instances underline the potential of docking-based AL models in drug discovery. By leveraging the predictive capabilities of docking scores within AL frameworks, researchers can enhance the drug discovery pipeline's speed and efficacy. This method not only improves the prioritization of therapeutic compounds but also promises broader applications in developing novel drugs for various diseases.”

- Line 473, the correct is ‘predictive affinity’ since this is a regression model using Gibbs free energy provided by docking or experimental Ki/Kd. Activity is for a classification model (not for regression model), in which a compound is active or inactive. Check this in other parts of the manuscript as well, please.

Response: Thank you for your comments concerning on the use of "predictive affinity" versus "activity." We appreciate your perspective and the distinction you've highlighted between regression and classification models within the context of molecular modelling. However, we would like to respectfully disagree with the suggestion to modify the terminology from "activity" to "predictive affinity". In the literature, the term "activity" is frequently used in a broader sense to describe the efficacy of a compound, encompassing both its binding affinity and its functional outcome at a molecular or cellular level. While it's true that "activity" is commonly associated with classification models where compounds are categorized as active or inactive, in the context of regression models predicting binding affinities, "activity" can still be appropriate. The term "activity" in the context of regression models is widely used in the literature to refer to the predicted or observed binding affinities of compounds (Peter et al., 2018). It encompasses the quantitative measure of how strongly a compound interacts with its target, whether derived from experimental data or computational predictions. 

We believe that our use of "activity" is consistent with established conventions in the field, where it often represents a continuum of molecular interactions rather than a binary classification. We have also reviewed other sections of our manuscript for consistency in terminology and found them to align with the broader scientific usage as supported by our references.

We hope this clarification addresses your concerns, and we thank you for prompting a thorough review of our terminology. We remain committed to accurate and clear scientific communication

-Peter, S. C., Dhanjal, J. K., Malik, V., Radhakrishnan, N., Jayakanthan, M., & Sundar, D. 2018. Quantitative Structure-Activity Relationship (QSAR): Modeling Approaches to Biological Applications. Encyclopedia of Bioinformatics and Computational Biology, 661-676. https://doi.org/10.1016/B978-0-12-809633-8.20197-0.)

. 

- In Figure 4, the X and Y axis labels should be changed to Affinity instead of Activity.

Response: Thank you for your observation. We understand your suggestion to change the labels from "Activity" to "Affinity". However, in the context of our study, at this stage the glide docking process employed was aimed at discriminate between binders and non-binders rather than quantifying the exact affinity of the binding interactions. As such, "Activity" as used here broadly categorizes the ability of compounds to interact with the target, rather than their precise binding affinities. Additionally, the labels on the X and Y axes are auto-generated by the program. We opted to retain the original output labels to maintain consistency and transparency in our data presentation.

We appreciate your attention to the details of our figure presentation and hope this explanation clarifies the terminology used in our study.

- Results sections – ‘induced fit docking’ - An image of the best novel and predicted compounds in the active site with the intermolecular interactions with the target should be provided.

Response: Thank you for the recommendation. An image of these compounds docked within the binding pocket has been provided.

- Table 2: What is the energy unit in Table 2? kcal/mol? In addition, Table 2 must be adapted to fit on the page.

Response: Thank you for your inquiry regarding the energy unit in Table 2 and the formatting of the table. We confirm that the energy unit in Table 2 is indeed k

---

## [Decision Letter · Decision Letter 1]

5 Aug 2024

Auto QSAR-based Active learning docking for hit identification of potential inhibitors of Plasmodium falciparum Hsp90 as antimalarial agents

PONE-D-24-01107R1

Dear Dr. Mokoena,

We’re pleased to inform you that your manuscript has been judged scientifically suitable for publication and will be formally accepted for publication once it meets all outstanding technical requirements.

Kind regards,

Yash Gupta, Ph.D.

Academic Editor

PLOS ONE

Additional Editor Comments (optional):

I think authors should add a limitation to the study stating mmGBSA and induced fit docking which consider target site to be flexible add too much variability for the used machine learning model.

Reviewers' comments:

Reviewer's Responses to Questions

**Comments to the Author**

1. If the authors have adequately addressed your comments raised in a previous round of review and you feel that this manuscript is now acceptable for publication, you may indicate that here to bypass the “Comments to the Author” section, enter your conflict of interest statement in the “Confidential to Editor” section, and submit your "Accept" recommendation.

Reviewer #2: All comments have been addressed

Reviewer #3: (No Response)

2. Is the manuscript technically sound, and do the data support the conclusions?

Reviewer #2: Yes

Reviewer #3: Yes

3. Has the statistical analysis been performed appropriately and rigorously? 

Reviewer #2: Yes

Reviewer #3: Yes

4. Have the authors made all data underlying the findings in their manuscript fully available?

Reviewer #2: Yes

Reviewer #3: Yes

5. Is the manuscript presented in an intelligible fashion and written in standard English?

Reviewer #2: Yes

Reviewer #3: Yes

6. Review Comments to the Author

Reviewer #2: Authors made appropriate corrections/explanations to the requested changes in the previous round. I do not have any objection to the manuscript.

Reviewer #3: Summary: The manuscript by Mokoena et al. presents an innovative approach combining Auto-QSAR models with active learning (AL), a type of machine learning, and molecular docking methods to identify potential inhibitors of Plasmodium falciparum Hsp90 (PfHsp90), a promising target for antimalarial drug development. The methods are detailed and well-structured, and the study successfully identifies several compounds with moderate activity against PfHsp90 and provides a starting point for further research and discovery.

Comment 1: The integration of Auto-QSAR, active learning, and docking is a commendable approach that enhances the efficiency of hit identification. The use of various computational techniques, including induced fit docking, molecular dynamics simulations, and MM-GBSA calculations, strengthens the reliability of the findings.

Comment 2: Some sections of the manuscript could be more concise to improve readability. The authors should emphasize the novelty and potential impact of the identified compounds on malaria treatment would strengthen this section, as well as eliminate any redundant information

Comment 3: The authors may have already addressed this, but the manuscript would benefit from ensuring all images are of the highest resolution to facilitate better comprehension of the results.

Concluding Remarks: I mostly agree with the previous reviewers. The manuscript presents a thorough study with significant potential implications for antimalarial drug discovery via PfHsp90. With improvements in image quality, conciseness, and a more detailed SAR analysis, I believe this manuscript does have a scientific impact and does fit this journal. Once what I consider minor improvements are completed, I believe this should move forward for publication.

7. PLOS authors have the option to publish the peer review history of their article (what does this mean?). If published, this will include your full peer review and any attached files.

Reviewer #2: **Yes: **Richard Beteck

Reviewer #3: No

---

## [Editor Report · Acceptance letter]

8 Aug 2024

PONE-D-24-01107R1 

PLOS ONE

Dear Dr. Mokoena, 

I'm pleased to inform you that your manuscript has been deemed suitable for publication in PLOS ONE. Congratulations! Your manuscript is now being handed over to our production team.

Kind regards, 

on behalf of

Dr. Yash Gupta 

Academic Editor

PLOS ONE